# LightCue: An Innovative Far-Red Light Emitter for Locally Modifying the Spectral Cue in Outdoor Conditions with Global Consequences on Plant Architecture

**DOI:** 10.3390/plants10112483

**Published:** 2021-11-17

**Authors:** Alain Fortineau, Didier Combes, Céline Richard-Molard, Ela Frak, Alexandra Jullien

**Affiliations:** 1INRAE, AgroParisTech, UMR EcoSys, Université Paris-Saclay, 78850 Thiverval-Grignon, France; alain.fortineau@inrae.fr (A.F.); celine.richard-molard@inrae.fr (C.R.-M.); 2INRAE, UR P3F, 86600 Lusignan, France; Didier.combes@inrae.fr (D.C.); elzbieta.frak@inrae.fr (E.F.)

**Keywords:** R:FR ratio, photomorphogenesis, light emitting diode, oilseed rape, branching

## Abstract

Plasticity of plant architecture is a promising lever to increase crop resilience to biotic and abiotic damage. Among the main drivers of its regulation are the spectral signals which occur via photomorphogenesis processes. In particular, branching, one of the yield components, is responsive to photosynthetic photon flux density (PPFD) and to red to far-red ratio (R:FR), both signals whose effects are tricky to decorrelate in the field. Here, we developed a device consisting of far-red light emitting diode (LED) rings. It can reduce the R:FR ratio to 0.14 in the vicinity of an organ without changing the PPFD in outdoor high irradiance fluctuating conditions, which is a breakthrough as LEDs have been mostly used in non-fluctuant controlled conditions at low irradiance over short periods of time. Applied at the base of rapeseed stems during the whole bolting-reproductive phase, LightCue induced an expected significant inhibitory effect on two basal targeted axillary buds and a strong unexpected stimulatory effect on the overall plant aerial architecture. It increased shoot/root ratio while not modifying the carbon balance. LightCue therefore represents a promising device for progress in the understanding of light signal regulation in the field.

## 1. Introduction

Architectural plasticity allows plants to adapt to or escape from environmental constraints, lengthening their stems to gain better access to light in competitive situations, or compensating for organ losses due to biotic or abiotic damage. Selecting plasticity traits adapted to different environmental constraints will enhance crop resilience and help reduce agriculture’s dependence on synthetic inputs [1,2]. Among the many architectural traits, branching plays a particular role as it is a major component of yield, determining the number of spikes or inflorescences. The plasticity (nature, sensibility, frequency, amplitude) of this process is high, driven by fluctuating environmental cues that trigger or inhibit bud development. Therefore, branching plasticity is a lever to compensate for damage caused by herbivore attacks, [3,4] but also to adapt to variations in crop density induced by agronomic practices, as shown for maize [5], wheat [6,7] and oilseed rape [4].

Whether a bud develops or not depends on entangled trophic, hormonal and environmental factors [8]. Among these factors, the light cues and particularly the red to far red ratio (R:FR) [1,8] are some of the most important signals influencing the architectural development of plants. The response involves several phytochromes with different light sensitivities that control photomorphogenesis [9]. The integration of processes at the whole-plant scale is, for example, known as shade avoidance syndrome (SAS) [9,10,11,12,13,14]. Typically, a decrease in the incident R:FR ratio below the value of one induces an increase in the rate of petiole and stem elongation and inhibits branching [1,12,13,15,16]. In dense crops under outdoor conditions, the spectral composition of the light environment under the canopy is heterogeneous and fluctuates depending, respectively, on the depth within the canopy [6] and the evolution of plant architecture [1,13]. The flux density of photosynthetically active photons (PPFD 400–700 nm), including the red (R) radiation (600–700 nm), is indeed mainly absorbed by the photosynthetic pigments and therefore decreases from the top to the bottom of the canopy, while FR (700–800 nm), which is not part of the PPFD spectral waveband, is mainly reflected and transmitted by plant organs and thus decreases at a slower rate with depth in the canopy. The relative differences in R and FR variations in the canopy thus leads to a decrease in the R:FR ratio with depth within the canopy [17]. Under agronomic densities, the basal axillary buds, being in both low PPFD and low R:FR environments, are generally inhibited, and plant architecture is that described in the SAS with an elongated and poorly branched stem. In contrast, the same plants grown under isolated plant conditions show a bushy architecture. However, because of the correlation between R:FR and PPFD within the canopy, it is not easy to conclude on the inhibitory role of a low R:FR per se on the number of branches. Although an inhibiting effect of the R:FR has been shown on wheat tillering by temporarily enriching the FR radiation on potted plants with tungsten lamps equipped with low-pass and high-pass filters [18]. Several other studies suggest an interaction between FR and PPFD. For example, the cessation of tillering in cereals has been linked to different local thresholds combining PPFD and R:FR levels and varying with genotypes and plant density [6,19,20]. For grassland species, response norms have been calibrated in controlled conditions, linking tillering delay, R:FR and PPFD [21,22]. In addition, the inhibiting effect of R:FR on axillary buds seem to vary according to species [14] and to developmental stage [9]. For instance, an early reduction in R:FR applied from the rosette stage on pot grown rapeseed plants led to an increased number of branches in oilseed rape [23], while it has no effect when applied from flowering. In the end, in outdoor conditions, and all the more in a dense field, if and when branching will develop is still not of clear cut.

The inhibition of axillary buds will affect the allocation of assimilates within the plant and thus impact the growth of the other organs (internodes, other branches). Indeed, many studies applying R:FR treatment at the whole plant scale shed light on the pleiotropy of the signal on plant growth and development and the interaction between local effect and transmitted distant effect: leaf elongation and tillering [24], internode elongation [25], signal transmission from shoot to roots and modification of above-ground/root ratios [26,27,28]. The study of these rearrangements and their quantitative evaluation are important elements in the understanding of plant plasticity and its effectiveness in terms of resilience. Thus, there are still too many knowledge gaps to properly explain the plasticity of plant branching in the field. Among them: first, which limiting factor prevails between R:FR and PPFD and according to which temporality? Second, what are the consequences of the local effect of the signal at the scale of the whole plant and at the scale of the growth cycle?

To address these questions, it is necessary to be able to isolate a specific signal on a given organ and evaluate its effect on its development as well as on the whole plant development and during the entire growth cycle. This requires overcoming of a major technical issue in the field, which is to be able to locally apply an FR signal sufficiently strong to significantly modify the R:FR ratio in a heterogeneous and fluctuating outdoor light environment, without inducing other modifications to the incident light, to avoid any interaction effect with the PPFD. Different methods using a variety of tools (lamps, filters, mirrors) have been developed to study the effect of R:FR on branching mostly indoors and less often outdoors. However, these methods have clear limitations for understanding the branching process. When carried out outdoors, even if they can consider long periods of growth, these methods often have the limitation of modifying both the R:FR and the PPFD [29], or of mobilizing lamp and filter combinations that are not adapted to a very fine localized application [23,30,31,32]. In contrast, by using LEDs or optic fibers [24,33,34,35], studies carried out under controlled conditions allow a finer control of the signal application and therefore manage to decipher the local effect of R:FR ratio according to the organ or tissue targeted. However, they do not deliver a sufficiently strong FR intensity modification to significantly affect the R:FR ratio under outdoor conditions characterized by high levels of radiation [34,35] and the results cannot be extrapolated to field conditions. Moreover, they are often carried out on plants at early vegetative stages and for short periods of time compared to the crop growth cycle. Due to all these technical limits, few studies have looked at the consequences of a local signal on the global architecture in the field and, as far as we know, none have considered the consequences of a triggered inhibition of the expansion of an axillary bud during the whole reproductive development.

We present here a new device called LightCue designed to apply a light treatment at the organ scale in outdoor conditions. We will answer two questions: first, does the local signal applied with LightCue to an axillary bud of plants grown outdoors, in isolated conditions without PPFD constraints, reproduce the global effects of R:FR signal widely documented in the literature on bud inhibition and stem elongation, but also on shoot/root biomass allocation? Second, does the integration of the responses to this local signal at the whole plant and the whole reproductive phase scales shed new light on the relationships between local light signals and the global regulation of architecture, in particular the dynamics of branching? Our results highlighted that the local R:FR: (1) was decreased below a threshold of 0.35 without affecting the local PPFD after installation of the LightCue device, and (2) the spectral signal modified the branching pattern as well as the overall architecture without changing the carbon balance other than the allocation between organs.

## 2. Materials and Methods

### 2.1. Plant Material and Growth Conditions

We chose to carry out the present study on winter oilseed rape (*Brassica napus* L.) as it is a very plastic and responsive species to SAS [4]. Oilseed rape is a dicotyledonous plant whose main stem is composed of a succession of phytomeres. Each of them is composed of an internode and a node bearing a leaf and an axillary bud. The total number of nodes on the main stem determines the branching potential of the plant. Each axillary bud may develop into a primary branch (expanded bud), or remain inhibited (non-expanded bud). Oilseed rape usually develops eight to ten apical branches when grown in dense crops while it can develop up to 35 to 40 apical branches when it is cultivated as an isolated plant, then showing a bushy shape. Since the total number of nodes is initiated and fixed during autumn and winter [36], an FR treatment applied after winter is expected to affect only stem elongation (internodes) and the expression of branching potential (number and length of branches).

On 4 March 2014, 18 winter oilseed rape plants (cv Kadore) were collected in the field with equivalent development stages according to their number of leaves and the size of their rosette (eight leaves and a rosette diameter of 12 to 15 cm). On 16 March 2014, before the stem elongation stage, 9 median plants out of the 18 ones were selected from phenological criteria and transplanted into 50 L pots filled with a mixture of silt loam soil and unfertilized potting soil (50/50). Plants were irrigated as needed to avoid hydric deficit and fertilized with the objective of expressing their growth potential. On 20 March, a first supply of 0.15 g of nitrogen (N) was applied in each pot. This dose corresponds to the nitrogen availability per plant in the field (50 kg N ha^−1^ for a density of 30 plants.m^−2^). On 16 April 2014, each pot was given a second supply of 0.33 gN (equivalent to 100 kg N.ha^−1^ for the same field density). The plants were grown outdoors, on a platform free of shade, five meters apart from each other to avoid competition for light.

### 2.2. LightCue Description

LightCue is a far-red (FR) emitter composed of three superimposed concentric rings each spaced 2.5 cm apart and made of transparent poly methyl methacrylate (PMMA) (Figure 1a,b). Each ring had an internal diameter of 5 cm and was equipped with five evenly distributed FR LEDs (ELD-740-354, Roithner Laser Technik, Vienna, Austria). The 40° emission angle of the LEDs ensures radiation coverage from half the radius to the center of the rings. The geometry of the LightCue emitter made it possible to irradiate a stem homogeneously and isotropically around its entire circumference and over a height of 5 cm.

The LED emission spectrum was measured with a spectroradiometer (UniSpec, PP-Systems, Amesbury, MA, USA) calibrated in irradiance within the 400–800 nm waveband, thereby including the spectral wavebands of PPFD (400–700 nm), red (R, (600–700 nm) and FR (700–800 nm). This spectroradiometer was equipped with an optic fiber linked to a cosine collector. The LED spectrum was centered on 760 nm (Appendix A). LEDs achieved a FR photon flux density (PFD) of 300 µmol.m^−2^.s^−1^ in the 700–800 nm spectral waveband at a distance of 2.5 cm (Appendix A). As this spectral waveband is beyond the visible, the LEDs did not impose any additional PPFD on the targeted organs.

The FR PFD from LightCue measured in the center of the rings on a horizontal surface is 100 µmol.m^−2^.s^−1^. As explained below (part 2.4), this value of 100 µmol.m^−2^.s^−1^ was used to calculate the effect of LightCue on the local R:FR.

FR LEDs emitted 15.7 W.m^−^² which represents 4–5% of the atmospheric radiation flux at our latitude. It was therefore considered that the energy input provided by LightCue was negligible and had no influence on the heating of the irradiated organs during the daytime period.

### 2.3. Spectral Treatments

We selected 5 control and 4 treated plants from the 9 median plants. Four plants in the so-called FR treatment were equipped with LightCue on the main stem to target the buds located at their base. For issues of space and speed of plant development, the installation of LightCue was phased over three weeks with one ring per week on 18 March, 25 March and 1 April. These devices remained in place during the stem elongation stage, for the duration of the experiment. During the experimental period, at Grignon (48°51′ N, 1°55′ E), the day lasts from 12:00 to 15:30. A timer controlled the lighting of the LED at dawn and the shutting down at dusk. In parallel, five plants in the so-called Control treatment were not equipped with rings.

### 2.4. Radiation and Spectral Transmittance Measurements

On 7 May 2014, at solar noon to minimize the effect of the position of the sun in the sky, the spectral measurements were made for each of the eight azimuths around the stem (N, NE, E, SE, S, SW, W, NW). The spectral distribution of the transmitted light around the stem was measured on three of the five Control plants with the spectroradiometer described above. Measurements were carried out with the collector positioned in the vicinity of the stem with its collecting surface horizontally facing upwards (yellow arrow on Figure 1b). All the data were analyzed using the Qua2Ray software [37]. During these spectral measurements, the incident PPFD was measured at 1 Hz with a quantum sensor (Li-190sz, LI-COR, Lincoln, NE, USA). The incident radiation measured above the plants and the transmitted radiation measured below the plants are used to calculate the percentage of transmitted radiation for each spectral waveband.

### 2.5. Estimation of the R:FR at the Base of the Plants Equipped with LightCue

Accurate and repeatable characterization of the LightCue contribution on the spectral environment of a plant organ is a challenge due to the fluctuating nature of this variable. In order to assess the daily variability of the applied spectral signal, the one-time transmitted light measurements taken on 7 May were extrapolated from the incident radiation data during a five-day period, from 5 to 9 May. It was considered that the structure of the plants changed very little over this period and therefore the self-shading conditions were stable. In addition, this period allowed us to take into account contrasting types of skies (a clear sky on 5 May and an overcast sky on the other days). From these measurements of transmitted light spectra and considering a constant incident light spectrum shape, we calculated quarter-hourly extrapolations of the ratio of transmitted red to incident red (Rt:Ri) and the ratio of transmitted FR to the incident FR (FRt:Fri). By applying these ratios to the incident PPFD, the time course of Rt and FRt was simulated on this five-day period. Finally, by adding the 100 µmol m^−2^ s^−1^ provided by LightCue to the simulated FRt, we obtained the quarter-hourly R:FR at the base of the FR treated plants. This estimated R:FR is considered as a proxy of the local spectral environment inside the FR-emitting rings, in the vicinity of the axillary buds.

### 2.6. Plant Measurements

#### 2.6.1. Non-Destructive Measurements

At regular intervals from the rosette to the flowering stages (16 March, 2 April, 11 April, 15 April and 28 April), the number of leaves, the node rank of developed branches and the plant height were recorded for each plant of each treatment. Node rank was recorded from the base to the top.

#### 2.6.2. Destructive Measurements

All the plants were collected and dissected on 15 May, in order to characterize the complete morphogenesis. On the main stem, the total number of nodes and their rank was deduced by counting the scars from the fallen leaves at the base of the stem. Each aerial organ of this axis (leaf, node, primary axillary bud and primary branch) was then numbered according to its rank from the base that corresponds to its order of appearance. The area of each individual leaf was measured with a LI-3100C area meter (Li-cor, Lincoln, NE, USA) as well as the length of each primary branch. On each primary branch, the total leaf area per branch was measured and the total number of secondary branches was counted. Finally, samples (main stem, individual leaves of the main stem, individual primary branch stems, and for each primary branch: pooled leaves of the branch, pooled secondary branch stems, pooled leaves of secondary branches, and the tap root) were dried for 48 h in an oven at 80 °C before their dry mass was measured.

### 2.7. Statistical Analysis

Given the low number of replicates (n = 4 and 5, for FR and Control treatments, respectively), mean comparison of the variables was performed using the Kruskal–Wallis test and the power of the tests was calculated using Anastat tools (www.anastat.fr, accessed on 1 December 2020). The comparison of organ number dynamics was performed using an ANOVA with repeated measurements with the R software (aov_ez, R Core Team 2020, R Foundation for Statistical Computing, Vienna, Austria, www.R-project.org/, accessed on 1 December 2020) considering the plants as the within-subject factor and the treatment as the between-subject factor. According to the results of the Maulchy’s sphericity test for the ANOVA with repeated measurements, the needed correction was made. This was the case for the test of the FR effect on the dynamics of the number of expanded buds (*p* > 0.29). The ANOVA with repeated measurements was also used to test the effect of the treatment on branch length considering the measurements recorded at each node rank of the plant as the repetition of the same measurement on a given subject.

To compare the branch length node by node in the two treatments, only nodes present and bearing a branch on at least three plants were considered. This was the case at the top of the plant, due to the variation in node numbers between plants: some ranks had only two repetitions. Similarly, at the bottom of the plant, the rank of the first basal branching varied. We only considered node ranks that had branches on at least three plants.

## 3. Results

### 3.1. Spectral Modification

The average R:FR measured was dramatically lowered in the FR treatment compared to the Control treatment (Figure 2), with R:FR = 0.17 ± 0.06 for the FR treatment versus R:FR = 0.56 ± 0.14 for the Control treatment. Meanwhile, there was no significant PPFD difference between FR and Control treatments. These results demonstrate the ability of our device to modify the R:FR ratio outdoor in a significant way without altering the PPFD.

### 3.2. R:FR Evolution

Simulations allowed us to explore the temporal variability around the mean value of the R:FR ratio within a single day and also from one day to another (Figure 3). Within a day, the R:FR ratio never exceeded 0.35. From one day to the next, two types of dynamics emerge: a very regular shape on 5 May, when the sky was clear, and clouds of points on the other days when the sky was cloudy. By construction, the dynamics of PPFD + FR follow the same pattern (see Appendix A).

The simulations showed that, whatever the incident PPFD, LightCue kept the spectral environment of the targeted buds significantly below an R:FR ratio of 0.35 all day long.

### 3.3. Plant Development and Architecture

The FR treatment modified the overall shape of the plants and their phenology (Figure 4). The plants illuminated at the base of their stem with LightCue were taller than the Control plants (Figure 5). The Control plants showed a bushy shape with one branch developed at the axil of each leaf, while the FR plants showed a more elongated shape with missing branches at the stem base, closer to the phenotype of plants in field conditions.

#### 3.3.1. Local Effect of the FR Signal on the Fate of Targeted Buds

The first visible effect caused by the FR signal was the inhibition of axillary buds located in the zone of influence of LightCue. Thus, while branching started as early as the fourth node for the Control plants, it did not start until the seventh node on all plants in the FR treatment (Figure 5). In both treatments, no branches developed from nodes 1 to 3. These nodes are short internodes at the base of the stem. They were only visible by the scars left by the fallen leaves at the base of the collar. At the time of collection, there was no difference in the number of nodes produced between FR-treated plants (14 ± 5) and Control plants (14 ± 7.3). However, in FR-treated plants, only 7.5 ± 1.1 nodes bore a branch between nodes 7 and 14, while, in Control plants, 9.6 ± 1.3 nodes bore a branch between nodes 5 and 14. As a result, the difference in the number of branches was significant (Figure 6). The analysis of the dynamics indicates that both treatments reached the maximum number of branches on 11 April and confirms the effect of the FR treatment (*p* < 0.0001).

#### 3.3.2. Associated Effects on the Extension of the Main Stem and Node Height

FR treatment increased the internodes length of the main stem (Figure 5), leading to higher nodes in the FR-treated plants compared to Control plants from node 11 to 13 (Kruskal–Wallis test *p* < 0.05). At the end of the experiment, node 13 was 40.0 ± 3.3 cm high for the FR treatment while it was 29.2 ± 3.9 cm high for the Control treatment, and FR-treated plants were significantly higher (91.5 ± 8.5 cm) compared to Control plants (72.3 ± 5.0 cm) (*p* < 0.05, power of tests > 70%).

#### 3.3.3. Associated Effects on Branch Development at the Plant Scale

For Control plants, final branch length increased from 47.0 ± 1.4 cm at node 5 to 67.5 ± 9.8 cm at node 10 (n ≥ 3, Figure 5). The length of the branches then decreased from node 10 to reach 40.7 ± 5.5 cm on node 14. For the FR-treated plants, the length of the branches increased from 46.7 ± 18.6 cm to 64.0 ± 10.4 cm between nodes 7 and 10 and then decreased to 56.0 ± 12.8 cm on node 13 (n ≥ 3). For both treatments, the maximal branch length was not significantly different and was similarly located on the 10th node. When comparing branch lengths for a given node rank, there was no difference between the treatments. However, when compared at a given height (independently of node rank), two parts can be distinguished on the stem. Below 20 cm, there was no significant difference in the branch length between the two treatments, while, above 20 cm, the branches of the FR treatment were significantly longer.

In summary, Control plants had a shrubby habit: they were short and developed a branch at each node of the main stem; the gradient in the length of these branches between the base and the top of the plant was dramatic. In contrast, the plants in the FR treatment had an elongated habit: they were tall and did not develop branches in the basal position, and the length gradient of these branches between the base and the top of the plant was low.

### 3.4. Global Effect on Carbon Economy and Allocation within the Plant

The FR treatment had no significant impact on the total leaf area (0.7 ± 0.2 m^2^ in both treatments, Figure 7a), nor on the total dry mass of the plants (138.1 ± 20.7 g for Control plants and 120.1 ± 10.1 g for FR-treated plants, Figure 7a). In contrast, the FR treatment significantly reduced the dry mass of the tap root (Figure 7c) and increased the shoot/root ratio from 7.7 ± 2.1 to 13 ± 4.3 (Figure 7d), indicating a re-allocation of carbon assimilates from the roots to the aerial part of the FR-treated plants. Within the above-ground part, no difference was found in leaf/stem ratio (1.1 ± 0.8 for the FR-treated plants and 1.3 ± 0.4 for the Control plants).

At the leaf scale (Figure 8), leaf area and the leaf mass per area (LMA) showed no significant differences between the two treatments.

## 4. Discussion

Our challenge was to reduce the R:FR ratio in the vicinity of axillary buds in outdoor conditions without affecting the PPFD to induce a modification of the branching process. Our results show that we achieved this goal, as local FR irradiation with LightCue emitters reduced the R:FR ratio in the immediate vicinity of the axillary buds by more than a factor of three, without altering the PPFD. A decrease in the R:FR ratio at the base of the plants during the stem elongation period significantly inhibited the expansion of the targeted buds (two branches less). This result, in addition to the elongation of the nodes of the main stem and the increase in the shoot/root ratio, is consistent with those widely highlighted in the literature but mainly obtained in controlled conditions, i.e., under low levels of PPFD and therefore not applicable to the field conditions [24,33,34,35]. However, this consistency validates the LightCue device as usable in outdoor conditions without controlling for other environmental factors, especially PPFD.

In addition, thanks to LightCue, this study provided new insights into the link between the local signal and the global effect at the plant scale. Indeed, we showed that inhibition of the targeted axillary buds had implications for both branching dynamics and stem elongation far beyond the targeted buds alone. The local signal applied was indeed able to generate a complete SAS on an isolated plant in outdoor conditions. Moreover, our results suggested that theses induced changes in the overall plant architecture were made at constant carbon balance, since neither leaf area nor plant dry matter were modified by the low R:FR treatment.

### 4.1. Controlling the Spectral Environment with LightCue

With the aim of modifying the branching process in outdoor conditions, one of the goals of this study was to develop a device capable of generating a sufficiently low R:FR signal throughout the day, regardless of the level, evolution, and fluctuations of the incident light in outdoor conditions, over an extended period of two months during which the branches unfolded. Given the number of spectroradiometers that would have been required, verifying that this objective was achieved would have been difficult through direct measurements. This is why we developed a method based on extrapolating the radiation transmitted under the plants. The simulated R:FR values for plants equipped with LightCue were consistent with the literature values measured at the bottom of the canopy in dense crops [6,20].

Most of the time, the methods presented in the literature for controlling the spectral signals do not only act on the spectral composition but also on the quantity of the incident light, especially when filters are used [23,29]. The use of LightCue emitters is a significant advance, as it made it possible to apply a spectral signal to a specific zone of the plant with quasi-isotropic FR sources without altering the local PPFD.

LEDs have already been used under controlled conditions in several studies but under extremely low and stable light conditions that are not representative of fluctuating outdoor conditions. For instance, PPFD only reached 125 µmol.m^−2^.s^−1^ [35] while natural light varies between 0 and 2000 µmol.m^−2^.s^−1^, with an R:FR ranging from 2.5 to 4.5 while the R:FR in natural light is about 1. Such controlled conditions make it possible to highlight isolate processes but not to understand their regulation in situ.

Phytochromes modulating the reversible photo-control of plants in response to low fluence (LFR, PFD < 1000 µmol m^−2^ s^−1^) also have an action in response to high irradiance (HIR) [38,39] which are not photo-reversible. These responses involving the same photoreceptors are different and even antagonistic, which supports the hypothesis of a strong interaction between them. In our study, there is a gradient of light conditions between HIR at the top of the plant (Appendix A) and LFR at the base of the stem. During plant development, the light environment of the buds evolves, which may induce an evolution of the type of response to light. Moreover, this gradient and the differential sensitivity to FR as a function of irradiance are likely to explain the FR–PPFD interactions identified in the literature [6,19,20,21,22]. This confirms the relevance of the study of responses in an outdoor environment with fluctuating radiation in spectral composition and intensity, much more complex than in controlled conditions.

Measuring the effect of the applied signal on the spectral environment of a bud is challenging, because, due to its nearly hemispherical shape, the bud collects radiation over its entire surface and therefore it is impossible to accurately measure the spectral signal it perceives. This is why we characterized the spectral environment in the vicinity of the buds and used this variable as a proxy of the environment they actually perceived. We could also have simulated the radiation intercepted by using a 3D plant model [40]. However, in our view, this approach seemed difficult to implement because of the complexity of the rapeseed architecture and the associated error in the quality of the 3D representation. The method we developed with LightCue emitters has the advantage of being accurate, repeatable, simple to carry out and complementary to the modelling approaches. However, this method assumes that the proportion of R and FR were stable within incident solar radiation during the day. In reality, the proportion of FR is greater at dawn and dusk. Therefore, by extrapolating the R and FR values from a series of measurements made on a single day at noon, the simulated R:FR values are slightly overestimated (with FR values being underestimated).

### 4.2. Modifying Shoot Branching with a Local R:FR Signal, from Axillary Bud to Whole Plant Architecture

Our results showed that FR treatment had a local inhibiting effect on bud expansion in the LightCue spectral footprint. The inhibiting effect of a low R:FR ratio on the expansion of basal buds that we highlighted in oilseed rape is consistent with the results from the literature on bud development under controlled conditions [14,41] and in the field where low R:FR ratios have been shown to reduce tillering or branching in wheat [6,19], soybean [42] and maize [32].

To our knowledge, there was only one specific study on the effect of R:FR on oilseed rape branching [23]. In this study, R:FR was modified globally by filtering the incident light applied over the entire growth cycle on isolated plants. The low R:FR treatment significantly increased the number of branches without any effect on above-ground biomass or yield. The effect on branching disappeared when the R:FR treatment was only applied from flowering onwards. These results illustrate the variability of plant responses depending on the method and timing of application of the treatment used. The authors hypothesize that the increase in branching with the early treatment was due to both a suppression of apical dominance, because the plants were grown in pots in isolated situations, and a higher remobilization of soluble carbohydrates from stems. However, this positive effect on branching was not observed when FR was added since flowering when the plants were also grown under isolated conditions, which is difficult to explain. The study does not specify whether the increase in branching was due to an increase in the number of nodes bearing an axillary bud on the main stem, i.e., an increase in branching potential, or to an increase in the number of expanded buds for the same total number of nodes. Indeed, it is also possible that the low R:FR treatment during the vegetative phase increased elongation of the main stem as well as the number of initiated nodes and thus the potential number of branches independently of the branching regulation during the stem elongation stage. Our study could help to refine the analysis of this global plant architectural response, by analyzing, for the first time, the length profile of the branches according to their position on the main stem and its response to a low R:FR signal. Results clearly show that the negative effect of a low R:FR ratio on the number of branches is due to a reduction in the branching rate with no effect on the total number of nodes and with an increased internode elongation.

Due to the inhibition of the basal axillary buds, each node above the footprint of the FR treatment is located higher on the stem than in the control treatment. These effects lead to a greater branch length at the top of the FR-treated plants compared to the controls, even if the largest branch was carried by the same node rank for both treatments. It seems that the branching pattern was pushed upwards, which is very similar to the SAS described in the literature [1,12]. These results confirm that the SAS we were able to induce is due to the effect of a local R:FR signal and that this signal acts even in the case of isolated plants, i.e., not subject to competition for light.

Our results show that the R:FR ratio applied locally on the FR-treated plants remained under a daily average of 0.17. This value was far below the threshold that induced a cessation of tillering in wheat (0.35 < R:FR < 0.40) [6]. These results are also coherent with the threshold values ranging from 0.18–0.22 for wheat lines with a tillering inhibition gene to 0.09–0.11 for free tillering wheat lines [20]. However, these thresholds were also associated with different values of transmitted PPFD. To the best of our knowledge, no threshold values inhibiting branching expansion have been reported in the literature in the case of oilseed rape so far.

### 4.3. Downstream Repercussions of Branching Modification 

The third main result of our study is that the local signal at the base of the plants had an impact not only on the targeted buds but also on the overall shoot architecture since plant height and carbon allocation were strongly affected.

While most studies in the literature present how the modification of the global environment of the plants and the integration of all signals at the whole plant scale can induce a local regulation of bud expansion, our study takes the opposite point of view by showing that a very local signal near a bud can affect both the bud fate and the global architecture of the plant. In addition, considering that each isolated plant had an equivalent total leaf area in both treatments, i.e., an almost equivalent carbon acquisition capacity, one possibility for the plant to respond to the spectral signal was to change its architecture by reallocating its assimilates in a different way. This is consistent with our observation of an increase in the shoot/root ratio for plants in the low R:FR treatment. It is also consistent with several studies carried out on various species grown in growing chambers or in the field, in monoculture or intercropping with different R:FR modification devices and which conclude that a low R:FR signal increased the shoot/root ratio. This increase in the shoot/root ratio could indicate a reallocation of assimilates to the aerial parts [43,44,45] even if there could be other modifications in the carbon budget, e.g., rhizodeposition, respiration, that we did not measure in our experiment. However, these results are consistent with recent work explaining the effect of R:FR when applied to above-ground parts on phytochrome- and auxin-mediated root growth, explaining a regulation of shoot/root ratios in response to light by hormones beyond an indirect effect by variation in assimilate availability and allocation [8,9,26,27].

It should also be noted that other architecture components not studied here, such as root architecture, may have been modified. For roots, modification of assimilate allocation could have been done either without modifying architectural features (homothetic reduction) or by modifying architectural traits such as the ratio of fine to suberized roots, or root segment diameters.

## 5. Conclusions

We demonstrated that the LightCue device we developed was relevant for controlling the spectral cue in the vicinity of targeted plant organs in outdoor light-fluctuating conditions. This device allowed achievement of a very low level of R:FR by reaching 0.14, low enough to induce significant photomorphological reactions at high fluctuating irradiance. Indeed, we demonstrated the significant effect of the R:FR spectral signal induced by LightCue on carbon allocation within the plants (lower shoot/root ratio) and on plant architecture at both local (two basal branches less) and whole plant scales (larger plants, longer upper branches). We assume LightCue is a promising tool to study the effect of local spectral cues on plant photomorphogenesis; for example, to decorrelate the effects of R:FR, PPFD and sugars in the hormonal regulatory network. By adapting LightCue to blue or UV wavelengths, it could also be used to unravel the regulatory role of each specific light signal in plant growth, resource acquisition and allocation, but also in plant defense mechanisms [46]. Given the geometry of LightCue, it can be adapted to the spatial conformation of organs of various species (wheat ears, cereal tillers, fruits), the rings can also be enlarged to fit larger organs. Finally, due to its small size and reduced footprint, LightCue could be used to modify the light spectrum within a dense canopy to study crop plasticity as a function of spectral conditions. This could be particularly relevant for improving crop resilience in mixed crop situations where competition for light is crucial for crop growth and yield. 

## Figures and Tables

**Figure 1 plants-10-02483-f001:**
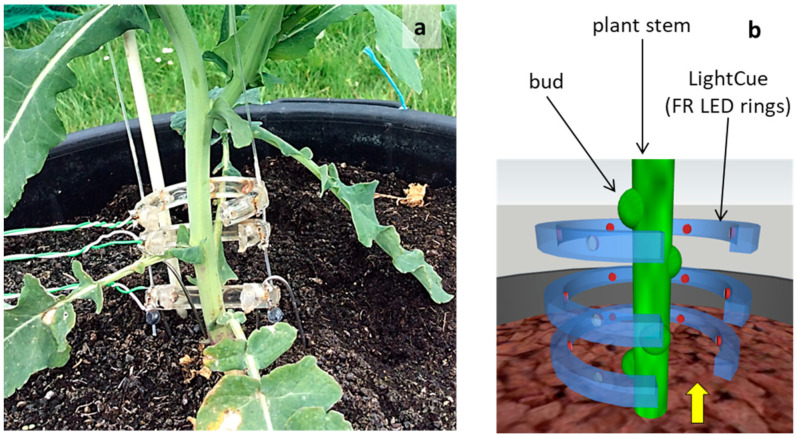
(**a**) Picture showing a plant equipped with LightCue (28 April 2014). (**b**) Diagram focusing on LightCue centered on the buds at the base of an isolated plant. The yellow arrow indicates the direction the spectroradiometer used to measure LED emission is pointing in.

**Figure 2 plants-10-02483-f002:**
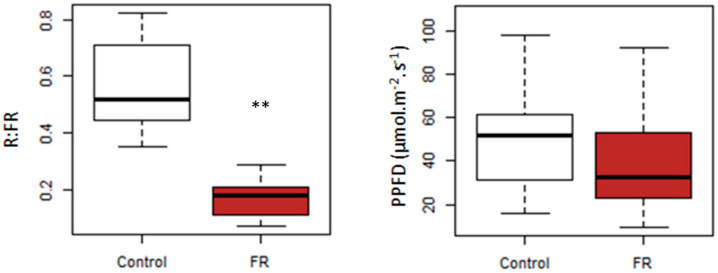
R:FR in Transmitted light (**left**) and PPFD in Transmitted light (**right**) measured at solar noon on 7 May at the stem base of the control plants (white) and the FR plants (brown). Kruskal–Wallis test, ** *p* < 0.01.

**Figure 3 plants-10-02483-f003:**
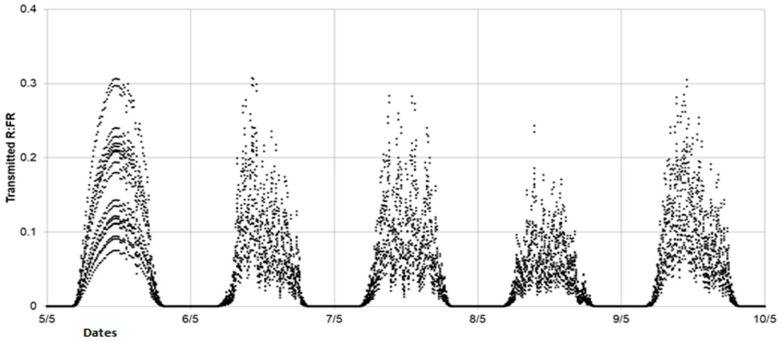
Quarter-hourly evolution of R:FR simulated at the base of the stem for 8 cardinal points around the FR plants.

**Figure 4 plants-10-02483-f004:**
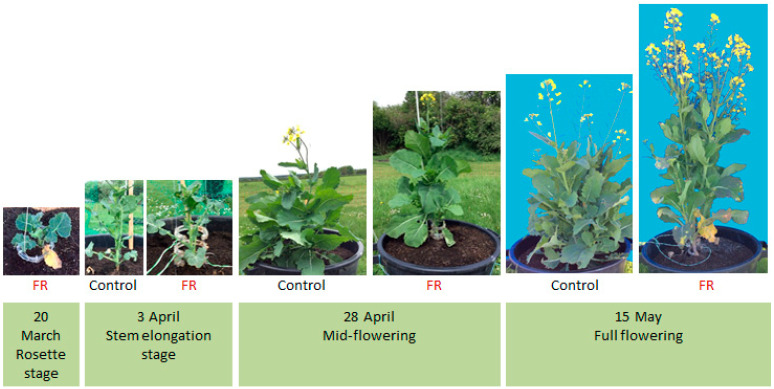
Development of Control- and FR-treated plants from the rosette stage (20 March 2014) to their full flowering stage (15 May 2014). The LightCue transmitters were added from 18 March to 1 April, ring by ring, up to three, as the stem grew.

**Figure 5 plants-10-02483-f005:**
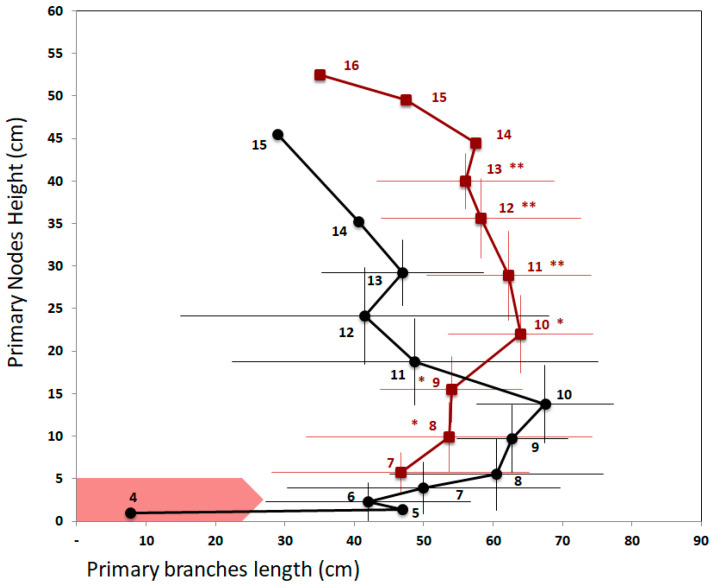
Profile of primary branch lengths according to node height on 15 May in Control (black) and FR-treated plants (brown). Numbers indicate the rank of the primary branch node. Horizontal and vertical bars represent the standard deviation calculated for n ≥ 3. The red arrow indicates the position of the FR-LED rings located between 0 and 5 cm. The node heights between Control and FR treatments were compared node by node using Kruskal–Wallis test: ** *p* < 0.05; * 0.05 ≤ *p*
*≤* 0.10. Branches of the Control plants were no longer than those of the FR-treated plants except above 20 cm height (Student test: *p* = 0.0032).

**Figure 6 plants-10-02483-f006:**
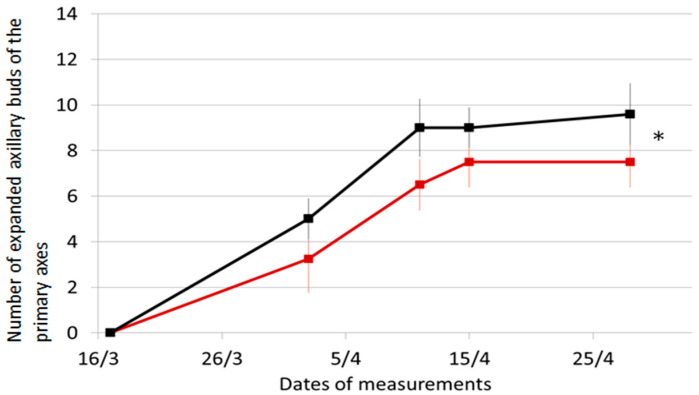
Evolution of the number of expanded primary axillary buds on Control—(black) and FR-plants—(brown). Vertical bars represent standard error of the mean with n = 4 or 5. Treatment effect * *p* < 0.0001, no interaction between treatment and dates of measurements was detected. The LightCue device was added on the 18 March.

**Figure 7 plants-10-02483-f007:**
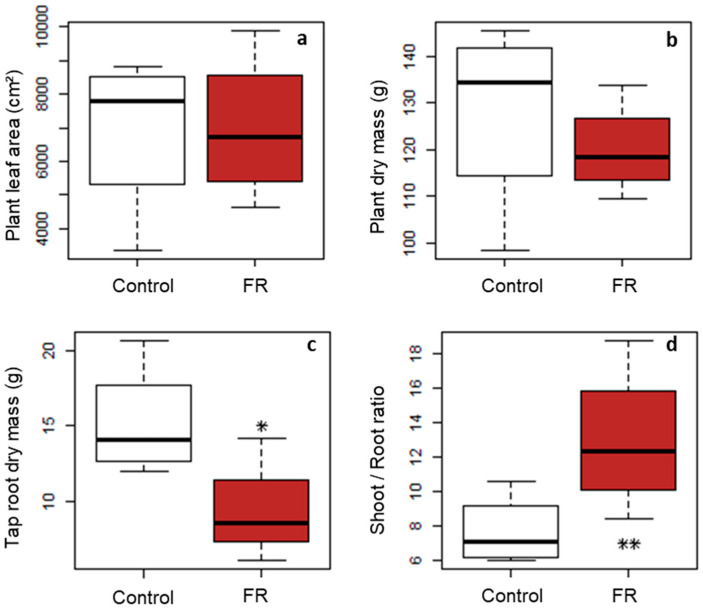
Effect of a local low far-red signal on (**a**) plant leaf area, (**b**) plant dry mass, (**c**) tap root dry mass and (**d**) shoot/root ratio measured at the time of collection, on 15 May on Control (white) and FR-treated plants (brown). Kruskal–Wallis tests (n = 4 or 5), * *p* < 0.10; ** *p* < 0.05. Powers of tests are 74% for the tap root (**c**) and 71% for the shoot/root ratio (**d**).

**Figure 8 plants-10-02483-f008:**
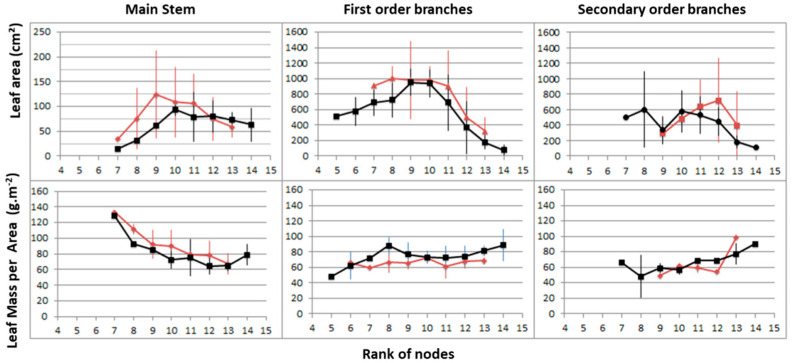
Pattern of variation in individual leaf area (upper part) and leaf mass per area (lower part) according to the node rank at the time of collection (15 May) in Control—(black) and FR-treated—(brown) plants (n = 3 or 4). Leaves of the main stem from nodes 1 to 4 had fallen before the day of collection.

## Data Availability

The data presented in this study are available in article and Appendix A.

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
