# Peer review of "LightCue: An Innovative Far-Red Light Emitter for Locally Modifying the Spectral Cue in Outdoor Conditions with Global Consequences on Plant Architecture"

_plants, 2021, doi:10.3390/plants10112483_

Round 1

Reviewer 1 Report

 Fortineau et al.  described the use of a LED FR device suitable for field studies which can locally alter R:FR ratio without significantly modifying the PPFD. Overall this study is sound an wall written.

The introduction is clear and well constructed, with very clear statement of the issue addressed in the study. The rationale is well explained. The authors show that previous devices designed to reduce R:FR ratio also modify the PPFD and/or are not suitable for long term application in field studies.

Results :

Using this device the authors could efficiently reduce R:FR ration locally at the base of tested plants, without altering the PPFD. Altough the number of replicates is low, the authors consistently obtained a difference in shape between test plants (elongated because of basal bud development inhibition up to node 7) and control plants (bushy, with basal node development inhibition up to node 5).

FR treatment induced  significant internode elongation, treated plants are taller.

One major effect of FR treatment at basal nodes was a strong reduction of root development with a much lower root dry mass as compared to control plants. The authors hypothesize that this is due to higher carbon reallocation towards the shoot.

I have two very minor remarks on this manuscript which is otherwise sound and clear.

  • The authors hypothesize that this is due to higher carbon reallocation towards the shoot. The authors could maybe undertone this statement as it might not be the unique explanation for dry mass difference.

It could be interesting to mention root architecture. How was root shape altered by FR treatment? Are root systems of bushy plants also bushy? Feedbacks from shoot with varied architectures to roots, which could explain impact on root development, could be of various nature, including hormonal or mechanical  feedbacks.

  •  In figure 1 :the authors provide a nice drawing of the LED device. I think it would be nice to combine with a  photo zooming on one ring so as to visualize the 4 LEDs’ position in real life.

Reviewer 2 Report

This work explains the effects of a light modulating device for the regulation of the plant response to red/far red wavelengths in an open environment. The novelty of this field of research is high if applied to crop production and vegetative growth regulation. The manuscript is properly structured and the results are well described, supporting the main conclusions. There is some information needed to be included that will help improving the fundament of the findings.

1) Introduction

- Please include the regulating roles of phytochromes on hormone-related mechanisms of plant development and reproduction events.

- Include the relevance of photoperiod and light intensity and quality in the development events.

- Line 48, please write what is PPFD before mentioning the abbreviation, just as with all the other abbreviations.

2) Materials and methods

- In the paragraph 2.3 Spectral treatments, please specify how many hours of FR light exposition were applied in the March-April season.

- In Figure 1, please add a photograph detailing how the real device looks installed in the shoot of the plant, analogue to Fig 1b.

3) Results: 

- How was the light intensity of sunlight + led light during the day? I suppose the lower intensity and maximum intensity described a gaussian shape. Do you have such measurements? maybe add it as a supplementary figure.

4) Discussion

- In section 3.2, please explain how R:FR will affect flowering tissue, knowing a flowering bud is a terminal fate of the tissue, sometimes linking reproduction with stress response, with an important redirection of nutrients and energy.

- Please consider and include how different are the plant responses under High Irradiance (more than 1000 µmol m-2 of light) where a non-FR reversibility phenomenon occurs, and under Low Fluence (under 1000 µmol m-2 of light) where FR response is reversible.

- Please explain how low R/FR regulates phytochromes, involved with auxin transport and cell expansion. Which can also help explaining why the roots presented reduced growth, in terms of auxin-dependent regulation of growth of underground tissues.

- Regarding the led light device, what kind of plants is this device appliable to? the shoot must fulfill some requirements?

- What developmental stages are best suitable for growth and development regulation? Are the responses going to be different for each developmental stage?

Reviewer 3 Report

The abstract should better represent the aim of the manuscript. Some numeric results should be added. Please reduce the number of keywords.

The introduction is too long. Authors should improve this part by adding only the most important information on the state of the art of this technique (technical information).

Materials&Methods should be reduced to better highlight the main information. Other information may be moved to the Supplementary materials.

Results and Discussion sections are very good and interesting but they should be integrated into the same paragraph to avoid redundant information.

Conclusions should be reported in order to summarize the main issues of the manuscript. It should be better linked to the main results highlighted in this study.
